# Initial presentation of early rheumatoid arthritis

**Lauri Weman**[1]*, **Henri Salo**[2], **Laura Kuusalo**[3], **Johanna Huhtakangas**[4], **Johanna Kärki**[5], **Paula Vähäsalo**[6], **Maria Backström**[7,8], **Tuulikki Sokka-Isler**[1]

**1** University of Eastern Finland and Jyväskylä Central Hospital, Jyväskylä, Finland, **2** Finnish Institute for Health and Welfare (THL), Data and Analytics, Helsinki, Finland, **3** Department of Internal Medicine, University of Turku and Turku University Hospital, Turku, Finland, **4** Division of Rheumatology, Kuopio University Hospital, Kuopio, Finland, **5** Department of Children and Adolescents, Kanta-Häme Central Hospital, Hämeenlinna, Finland, **6** Department of Children and Adolescents, Research Unit of Clinical Medicine, University of Oulu, Oulu University Hospital and Medical Research Center, Oulu University Hospital and University of Oulu, Oulu, Finland, **7** Department of Paediatrics, Wellbeing Services County of Ostrobothnia, Vaasa, Finland, **8** Research Unit of Clinical Medicine, University of Oulu, Oulu, Finland

* lauriweman@gmail.com

**Data Availability Statement:** The Quality Register data must be used strictly for the purposes of the Quality register, by the specialists nominated for this task, under the Finnish Institute for Health and Welfare, which is supervised by the Ministry of

## Abstract

### Objectives

To study the joint distribution and clinical picture of rheumatoid arthritis (RA) at the initial presentation in seropositive (anti-citrullinated protein antibody (ACPA) and/or rheumatoid factor (RF) positive) and negative patients and the effect of duration of symptoms on the clinical picture.

### Methods

Data of patients who received reimbursement for DMARDs for newly diagnosed RA in 1/2019 to 9/2021 were extracted from the national databases. Joint counts, presence of symmetrical swelling, other disease activity measures, and patient reported outcomes (PROs) were compared in seropositive and negative patients. Regression analyses were applied to compare clinical variables in patients with duration of symptoms of <3, 3–6, and >6 months, adjusted for age, sex, and seropositivity.

### Results

Data of 1816 ACPA and RF-tested patients were included. Symmetrical swelling was present in 75% of patients. Seronegative *versus* positive patients had higher value for all disease activity measures and PROs including median swollen joint count (SJC46 10 *versus* 5) and DAS28 (4.7 *versus* 3.7), (p<0.001). Patients diagnosed in <3 months had higher median pain VAS (62 *versus* 52 and 50, p<0.001) and HAQ (1.1 *versus* 0.9 and 0.75, p = 0.002) compared to those with a duration of symptoms of 3–6 and >6 months. Patients diagnosed >6 months were ACPA-positive more frequently (77% *versus* 70% in other groups, p = 0.045).

Social Affairs and Wealth. The Finnish Institute for Health and Welfare is the owner of the data. Queries concerning the quality register data can be directed to the institute FinData, which is mandated to grant approval to use anonymized quality register data for research purposes. E-mail: data@findata.fi.

**Funding:** We would like to thank The Finnish Psoriasis Association, The Finnish Society for Rheumatology, and Amgen. The author also acknowledges the Research Committee of the Kuopio University Hospital Catchment Area for the State Research Funding (project 5041730, Kuopio, Finland). The funders had no role in study design, data collection and analysis, decision to publish, or preparation of the manuscript.

**Competing interests:** The authors have declared that no competing interests exist.

## Conclusion

Incident RA presents mainly as symmetric arthritis. Seronegative patients have higher disease burden at the initial presentation. Patients experiencing more severe pain and decreased functional ability are diagnosed earlier, regardless of ACPA- status.

## Background

Before the introduction of modern diagnostic tools, incident rheumatoid arthritis (RA) was recognized as symmetric polyarthritis. Symmetry was one criterion in the previous 1987 classification criteria for RA [1], which were developed to differentiate RA from other types of inflammatory arthritides. Symmetry of joint erosions has been observed in patients with early RA diagnosed in the early 1990's [2] and in patients with established RA in 1963–1996 [2,3]. However, no studies have been conducted recently on the symmetry of joint involvement at the presentation of RA [4]. Although not part of the current classification criteria [5], symmetry of arthritis is still being taught in textbooks and medical schools, as one of the hallmarks of incident RA. However, considering RA only in cases with symmetric polyarthritis might postpone the suspicion of early RA in primary health care.

In the 2010 ACR/EULAR classification criteria for RA, more emphasis has been placed on serology [5] without requirement for symmetry. With the 2010 criteria, early RA is now presented mainly in two forms with 6 criteria fulfilled, either seropositive RA, which can be oligoarthritis without elevated inflammatory markers, or seronegative RA, which is usually polyarthritis with elevated inflammatory markers. In fact, it is getting obvious that autoantibodies are the hallmark of RA and seronegative RA is a distinct entity [6] with different risk factors [7,8] and outcomes compared to seropositive RA [9]. Our recent study indicated that seronegative RA turns out to be a spectrum of different conditions when observed for 10 years [10,11]. Nonetheless, seronegative active arthritis requires early identification and tightly controlled treatment with Disease Modifying Antirheumatic Drugs (DMARDs) as it also causes significant treatable disease burden to patients [12].

First in the 1970's [13] and more widely since the 1990's, it has been recognized that RA should be identified and its treatment initiated as soon as possible to facilitate good outcomes [13,14]. With multiple studies showing that early recognition is crucial, it has since been emphasized as an important approach for the management of RA [15]. It has also been shown that treatment initiation within the first 12 weeks after the first symptoms might be the best window of opportunity in terms of treatment outcomes [16,17]. Some studies suggest that the duration of symptoms has decreased during the current millennium [18]. However, there are still notable delays in terms of treatment initiation. For example, only 31% of patients with incident RA were examined by a rheumatologist within 12 weeks of the first symptoms in the Netherlands in 1993–2006 [19]. In the UK, patients with early RA visit their general practitioner 4 times on average before being referred to a specialist according to a report from 2008–2009 [20].

Therefore, our objective was to study the initial presentation of RA in a nationwide setting, with a focus on the serological status and symmetry of joint swelling, as well as the effect of duration of symptoms.

## Methods

### Setting

Patients with incident RA are diagnosed and treated in rheumatology outpatient clinics in Finland. According to the national guidelines, DMARDs are started at the time of the diagnosis

together with a medication reimbursement application, prepared by the rheumatologist. The reimbursement is granted by the Social Insurance Institution of Finland (KELA) for DMARDs. In addition to granting reimbursements for patients with chronic illnesses, such as RA, KELA also maintains a database of the individuals, containing the ICD code of the diagnosis, date of the reimbursement and basic demographic data of the individuals.

For the current study, patients who received their first reimbursement for DMARDs prescribed for the treatment of RA between 1/2019-9/2021 were identified in the KELA database. Clinical and demographic data were extracted from The Finnish Rheumatology Quality Register, using the individual identification code. To capture the data at the time of the diagnosis, clinical data were extracted from the most recent visit within 0 to 90 days prior to the date of the medication reimbursement; visits occurred between 23[rd] November 2018 and 30[th] August 2021. All patients were over 16 years old and could not be diagnosed as any other specific arthritis. The ACR/EULAR 2010 classification criteria were used to aid in diagnosing the patients [5].

## Variables

Following variables were collected at the time of the diagnosis of RA and were available for analyses.

*Demographic variables*

- Age in years

- Sex; male subjects/female subjects

- Smoking status: current smoking; having ever smoked

- Employment status for patients under 65 years old; currently working, unemployed, disabled and not in work force such as home makers and students

  *Disease characteristics*

- Duration of symptoms

- Fulfillment of ACR/EULAR 2010-criteria [5].

  *Clinical variables*

- Swollen joint count (SJC) and Tender joint count (TJC) on 46 joint counts

- Distribution of swollen and tender joints on 46 joint counts

- Presence of symmetrical swelling in the MCPs, PIPs, wrists, MTPs, knees, ankles, and the elbows

- C-reactive protein (CRP), erythrocyte sedimentation rate (ESR) and DAS28 scores

- Investigator's global assessment of disease activity (Dr.global)

- Patient reported outcomes (PROs) such as pain, fatigue, and Patient Global Assessment (PGA)

- Self-reported functional capacity according to the Stanford Health Assessment Questionnaire (HAQ).

  *The Initial Medication*

- The medications of interest were methotrexate (MTX), other conventional synthetic disease-modifying antirheumatic drugs (csDMARDs), such as hydroxychloroquine (HCQ) and

sulfasalazine (SSZ), biological DMARDs (bDMARDs) and janus kinase (JAK) inhibitors, as well as systemic glucocorticoids (GC).

*Duration of symptoms* is the time between the first symptoms and the diagnosis of RA, recorded in months. At the initial visit, the patient was asked when he/she recognized the first symptoms of RA.

*ACPA- positivity* was defined by the laboratory reference values, which was $\geq$ 7 kU/l.

*RF-positivity* was defined by the laboratory reference values, being $\geq$ 15 IU/ml.

*Fulfilment of the ACR/EULAR 2010 criteria* was calculated by the examining physician, who counts the items in the ACR/EULAR 2010-criteria. If the total score adds up to 6 or more points, the ACR/EULAR 2010-criteria are met, when other reasons that can explain the condition are excluded [5].

**Joint counts.** The presence of swollen and tender joints was calculated using a 46 joint count including proximal interphalangeal (PIP), and metacarpophalangeal (MCP) joints, interphalangeal (IP) joints of the thumbs, wrists, metatarsophalangeal (MTP) joints, distal interphalangeal (DIP) joints of big toes, temporomandibular joints as well as ankle-, knee-, hip-, elbow- and shoulder joints.

*Symmetrical swelling* was analyzed from the MCPs, PIPs, wrists, MTPs, knees, ankles and the elbows, according to the 1987 criteria [1]. A patient was considered to have symmetrical swelling if he/she had bilateral swelling in the same anatomical site. Symmetrical joint tenderness wasn't analyzed as it wasn't in the 1987 criteria [1].

*CRP and ESR* were determined according to the laboratory reference values for women and men in different age groups. A reference value of below 10 mg/l was normal for CRP. An elevated ESR included values of $\geq$ 20 mm/h for women younger than 50 years, $\geq$ 30 mm/h for women over 50 years and $\geq$42 mm/h for women over 85 years and $\geq$ 15 mm/h for men younger than 50 years, $\geq$20 mm/h for men over 50 years and $\geq$30 mm/h for men over 85 years.

*DAS28* was used to describe disease activity [21].

*Dr.global* included the physicians' assessment of rheumatic activity on the 0–100 mm Visual Analog Scale (VAS), where 0 equals no disease activity and 100 maximal disease activity.

*PROs* included the self-assessment of pain, fatigue, and PGA on the 0–100 mm VAS, where 0 equals no symptoms and 100 maximal discomfort.

*HAQ* was used to describe the functional capacity of patients. It is scored from 0 to 3, a score of <0.5 is a sign of good functional status. In this study, we used HAQ without "aids and devices" due to its better accuracy in clinical studies [22,23].

*The Initial Medications* were analyzed in following groups:

- MTX monotherapy

- MTX in combination with (an)other csDMARD(s)

- csDMARD monotherapy or a combination

- bDMARD with or without MTX

- JAK-inhibitor with or without MTX

- GC, regardless of DMARDs

Clinical and demographic variables were determined for all patients and for seropositive and negative patients. Comparisons were conducted between seropositive and negative patients as well as between only ACPA-positive and only RF-positive patients with clinical variables. The use of medication was determined only for seropositive and seronegative patients and their use was compared between these groups.

### Methods to study the effect of duration of symptoms on disease activity

Three groups were formed according to the duration of symptoms before receiving the diagnosis: patients with a duration of symptoms of <3 months (group 1), patients with a duration of symptoms of 3–6 months (group 2) and patients with a duration of symptoms of over 6 months (group 3). SJC46, TJC46, serological status, CRP, ESR, DAS28, PROs, HAQ, and Dr. global were compared between these groups.

### Statistical methods

Descriptive statistics were used with mean values with standard deviation (SD) and median values with interquartile ranges (IQR) depending on the way a value is distributed. Chi-square test was used to compare categorical variables. P = 0.05 was set as a threshold for statistical significance. Regression models were used to compare clinical and demographic variables, ANOVA for continuous variables and logistic regression for dichotomous correlatives and for comparisons of median values transformed as median splits. Adjustments for age and sex were used in the comparisons of patients by ACPA-status. Variables were adjusted for seropositivity, age and sex in the comparisons of groups by duration of symptoms.

Analyses were conducted using the R Statistical language (version 4.2.1; R Core Team, 2022) on Ubuntu 20.04.5 LTS.

### Ethical issues

This study was conducted as a register-based study using data from the Finnish Rheumatology Quality Register, which is kept by the Finnish Institute for Health and Welfare (THL), which granted the approval for the study. In a register-based study, patient consent was not required.

## Results

### Demographic variables

A total of 2017 patients with incident RA were identified in the database. ACPA and/or RF were available for 1816 (90%) patients, which were included in the analyses. Of these, 1444 (80%) were seropositive and 372 (20%) were seronegative. A total of 5 (0.3%) seropositive patients weren't tested for ACPA, as they were already RF-positive. Vice-versa, 83 (6%) patients weren't tested for RF, as they already had a positive ACPA-result. The mean (SD) age was 59 (16) years for all patients. For seropositive and negative patients, the corresponding numbers were 59 (15) and 60 (16) (p<0.001). A total of 1034 (57%) patients overall, 850 (59%) seropositive patients and 184 (49%) seronegative patients were under 65 years old at the initial visit (p = 0.002) (Table 1).

A total of 19% of all patients were current smokers and 55% had ever smoked. The corresponding proportions were 21% and 57% for seropositive patients and 12% and 46% for seronegative patients (p<0.001 for both comparisons, adjusted for age and sex). In terms of employment status, a total of 69% of all patients were employed, 20% disabled, 8% unemployed and 4% weren't currently in work force. No major differences were found between seropositive and negative patients (Table 1).

### Disease characteristics

The median (IQR) duration of symptoms before being diagnosed was 4 (2, 10) for all patients. For seropositive patients it was 5 (2, 10) and 4 (2, 8) for seronegative patients (p = 0.030, adjusted for age and sex).

**Table 1. Demographic data and disease characteristics of patients with RA at the initial presentation, by serological status.**

| Variable | Available data for all patients | All | Available data for seropositive n, (%) | Seropositive | Available data for seronegative n, (%) | Seronegative | p-value |
|---|---|---|---|---|---|---|---|
| n | 1816 (100%) | 1816 | | 1444 (80%) | | 372 (20%) | |
| Females, n (%) | 1816 (100%) | 1177 (65%) | 1444 (100%) | 947 (65%) | 372 (100%) | 230 (62%) | |
| Mean (SD) Age in years<br>All | 1816 (100%) | 59 (16) | 1444 (100%) | 58 (16) | 372 (100%) | 60 (16) | <0.001 |
| Smoking status, n (%)<br>Current smokers<br>Patients that have ever smoked | 1707 (94%) | 321 (19%)<br>942 (55%) | 1363 (94%) | 281 (21%)<br>783 (57%) | 344 (92%) | 40 (12%)<br>159 (46%) | <0.001<br><0.001 |
| Patients under 65 years old, n (%) | | 1034 (57%) | | 850 (59%) | | 184 (49%) | 0.002 |
| Employment status for patients under 65 years old, n (%)<br>Employed<br>Disabled<br>Unemployed<br>Not in work force | 733 (71%) | 504 (69%)<br>148 (20%)<br>52 (8%)<br>29 (4%) | 601 (71%) | 415 (69%)<br>119 (20%)<br>46 (8%)<br>21 (3%) | 132 (72%) | 89 (67%)<br>29 (22%)<br>6 (5%)<br>8 (6%) | 0.273 |
| Disease characteristics | | | | | | | |
| Median (IQR) duration of symptoms in months | 1288 (71%) | 4 (2, 10) | 1026 (71%) | 5 (2, 10) | 262 (70%) | 4 (2, 8) | 0.030 |
| Proportions of patients fulfilling the ACR/EULAR 2010 criteria | 1497 (82%) | 1311 (88%) | 1165 (81%) | 1064 (91%) | 332 (89%) | 247 (74%) | <0.001 |

The proportions of patients fulfilling the ACR/EULAR 2010-criteria at the initial presentation were 88% for all patients, 91% for seropositive and 74% for seronegative patients (p<0.001, adjusted for age and sex) (Table 1).

## Swollen- and tender joint counts

The median (IQR) SJC46 was 6 (3, 10) for all patients. For seropositive patients it was 5 (2, 9) and 10 (6, 15) for seronegative patients (p<0.001). The values for TJC46 were 6 (3, 11) for all patients and 5 (2, 10) and 10 (5, 18) for seropositive and negative patients, respectively (p<0.001, adjusted for age and sex) (Table 2).

For all patients, 23% had two or less swollen joints at the initial presentation, 25% had 3–5 and 52% had ≥6 swollen joints on the 46 swollen joint count. The corresponding proportions were 26%, 28% and 46% for seropositive patients and 9%, 16% and 75% for seronegative patients (p<0.001, adjusted for age and sex) (Table 2).

A total of 23% of all patients had two or less tender joints in the 46 tender joint count, 25% had 3–5 and 52% ≥6 tender joints. For seropositive patients the proportions were 26%, 28% and 46% and 14%, 16% and 70% for seronegative patients (p<0.001, adjusted for age and sex) (Table 2).

## Pattern and symmetry of joint involvement at the initial presentation

At the initial presentation, wrists were the most commonly affected joints, with 43.5% of patients having a swollen left wrist and 43.6% a swollen right wrist. Any joint from the right MCP's was swollen in 9.5 to 34.9% of patients and in 9.7 to 28.7% of patients from the left MCP's. The corresponding proportions were 9.5 to 31.5% for the right PIP's and 8.2 to 25.4%

**Table 2. Clinical data of patients with RA at the initial presentation, by serological status.**

| Variable | Available data for all patients n, (%) | All | Available data for seropositive patients n, (%) | Seropositive | Available data for seronegative patients n, (%) | Seronegative | p-value |
|---|---|---|---|---|---|---|---|
| **Number of patients, n** | | 1816 | | 1444 (80%) | | 372 (20%) | |
| **Median (IQR) SJC46** | 1723 (95%) | 6 (3, 10) | 1371 (95%) | 5 (2, 9) | 352 (95%) | 10 (6, 15) | <0.001 |
| **Median (IQR) TJC46** | 1723 (95%) | 6 (3, 11) | 1371 (95%) | 5 (2, 10) | 352 (95%) | 10 (5, 18) | <0.001 |
| **Proportions of patients with different SJC46 n, %**<br>≤2<br>3–5<br>≥6 | | 394 (23%)<br>434 (25%)<br>894 (52%) | | 362 (26%)<br>339 (28%)<br>629 (46%) | | 32 (9%)<br>55 (16%)<br>265 (75%) | <0.001 |
| **Proportions of patients with different TJC46 n, %**<br>≤2<br>3–5<br>≥6 | | 404 (23%)<br>435 (25%)<br>884 (51%) | | 356 (26%)<br>378 (28%)<br>637 (46%) | | 46 (14%)<br>57 (16%)<br>247(70%) | 0.014 |
| **Proportions of patients with symmetrical swelling by anatomical site n, %**<br>Any site<br>PIPs<br>MCPs<br>Wrists<br>Elbows<br>Knees<br>Ankles<br>MTPs | 1624 (89%) | 1214 (75%)<br>537 (33%)<br>568 (35%)<br>550 (34%)<br>60 (4%)<br>193 (12%)<br>136 (8%)<br>567 (35%) | 1285 (89%) | 906 (71%)<br>375 (29%)<br>377 (29%)<br>365 (28%)<br>34 (3%)<br>124 (10%)<br>73 (6%)<br>435 (34%) | 339 (91%) | 308 (91%)<br>162 (48%)<br>191 (56%)<br>190 (56%)<br>26 (8%)<br>69 (20%)<br>63 (19%)<br>132 (39%) | <0.001<br><0.001<br><0.001<br><0.001<br><0.001<br><0.001<br><0.001<br>0.097 |
| **Median (IQR) CRP** | 1717 (95%) | 8 (3, 25) | 1361 (94%) | 7 (3, 20) | 356 (96%) | 19 (5, 46) | <0.001 |
| **Median (IQR) ESR** | 1536 (85%) | 22 (10, 39) | 1221 (85%) | 20 (10, 37) | 315 (85%) | 26 (12, 46) | 0.006 |
| **Proportions of patients with normal CRP or ESR n, %**<br>CRP<br>ESR | | 903 (53%)<br>929 (60%) | | 777 (57%)<br>768 (63%) | | 126 (35%)<br>191 (51%) | <0.001<br><0.001 |
| **Median (IQR) DAS28** | 1644 (91%) | 3.9 (3.1, 4.7) | 1308 (91%) | 3.7 (3.0, 4.5) | 336 (90%) | 4.7 (4.0, 5.7) | <0.001 |
| **Mean (SD) DAS28** | | 4.0 (1.2) | | 3.7 (1.1) | | 4.8 (1.3) | <0.001 |
| **Median (IQR) Dr.global** | 1543 (85%) | 40 (26, 56) | 1233 (85%) | 39 (25, 50) | 310 (83%) | 50 (38, 69) | <0.001 |
| **Median (IQR) PROs**<br>Pain<br>Fatigue<br>PGA | 1593 (88%)<br>1418 (78%)<br>1590 (88%) | 54 (30, 75)<br>40 (14, 69)<br>48 (23, 64)<br>45 (18, 70)<br>50 (25, 67) | 1272 (88%)<br>1128 (78%)<br>1273 (88%) | 51 (30, 74)<br>40 (14, 69)<br>48 (23, 64) | 321 (86%)<br>290 (78%)<br>317 (85%) | 61 (39, 77)<br>52 (31, 73)<br>54 (32, 70) | <0.001<br><0.001<br><0.001 |
| **Median (IQR) HAQ** | 1432 (79%) | 0.90 (0.50, 1.40) | 1144 (79%) | 0.88 (0.38, 1.4) | 288 (77%) | 1.10 (0.75, 1.50) | <0.001 |

*(Continued)*

**Table 2.** (Continued)

| Variable | Available data for all patients n, (%) | All | Available data for seropositive patients n, (%) | Seropositive | Available data for seronegative patients n, (%) | Seronegative | p-value |
|---|---|---|---|---|---|---|---|
| Mean (SD) HAQ | | 0.97 (0.69) | | 0.93 (0.69) | | 1.20 (0.66) | |
| Initial Medication | | | | | | | |
| MTX monotherapy | | | | 242 (17%) | | 86 (23%) | |
| MTX and csDMARD | | | | 1039 (72%) | | 249 (67%) | |
| only csDMARD | | | | 159 (11%) | | 36 (10%) | |
| bDMARD | | | | 3 (0%) | | 1 (0%) | |
| JAK-inhibitor | | | | 1 (0%) | | | |
| GC | | | | 1181 (82%) | | 348 (94%) | |

for the left PIP's and 10.3 to 28.0% for the right and 11.0 to 27.6% for the left MTPs. Knees were the most commonly swollen large joints (20.8% for right and 17.2% for the left). The corresponding proportions were 9.6% and 7.9% for shoulders, 13.5% and 12.8% for ankles, 6.8% and 5.9% for elbows and 1.9% and 2.2% for the hips (S1 Table).

At the initial presentation of RA, a total of 75% of all patients had symmetrical swelling in any of the anatomical sites. In different anatomical sites, 35% had symmetrical swelling in MCPs, 33% in the PIPs, 34% in wrists, 35% in the MTPs, 12% in the knees, 8% in the ankles and 4% in the elbows. A total of 71% of seropositive patients had any symmetrical swelling, and the corresponding proportion was 91% for seronegative patients ($<0.001$, adjusted for age and sex). In terms different anatomical sites, seronegative patients had more symmetrical swelling in all sites, with the biggest differences in the wrists compared to seropositive patients (28% in seropositive and 56% in seronegative patients) ($p<0.001$, adjusted for age and sex) and the MCPs (29% and 56%) ($p<0.001$, adjusted for age and sex). A total of 34% of seropositive patients had symmetrical swelling in the MTPs and 39% of seronegative patients ($p = 0.097$) (Table 2, Fig 1).

## Disease activity measures

The median (IQR) CRP was 8 (3, 25) for all patients with incident RA, 7 (3, 21) for seropositive and 15 (5, 46) for seronegative patients ($p<0.001$). The median (IQR) ESR was 22 (10, 39) for all patients 20 (10, 37) for seropositive and 26 (12, 46) for seronegative patients ($p = 0.006$, adjusted for age and sex).

A total of 53% of all patients had normal CRP and 60% normal ESR. For seropositive and negative patients, the proportions were 57% and 35% for CRP ($p<0.001$) and 63% and 51% for ESR, respectively ($p<0.001$, adjusted for age and sex) (Table 2).

At the initial presentation, the median (IQR) DAS28-scores were 3.9 (3.1, 4.7) for all patients, 3.7 (3.0, 4.4) for seropositive patients and 4.7 (4.0, 5.7) for seronegative patients ($p<0.001$, adjusted for age and sex) (Table 2).

For Dr.global, the median (IQR) value was 40 (26, 56) for all patients, 39 (25, 50) for seropositive patients and 50 (38, 69) for seronegative patients ($p<0.001$, adjusted for age and sex) (Table 2).

## Patient reported outcomes

For pain, the median (IQR) VAS-score was 54 (30, 75) for all patients, 51 (30, 74) for seropositive patients and 61 (39, 77) for seronegative patients ($p<0.001$, adjusted for age and sex). For

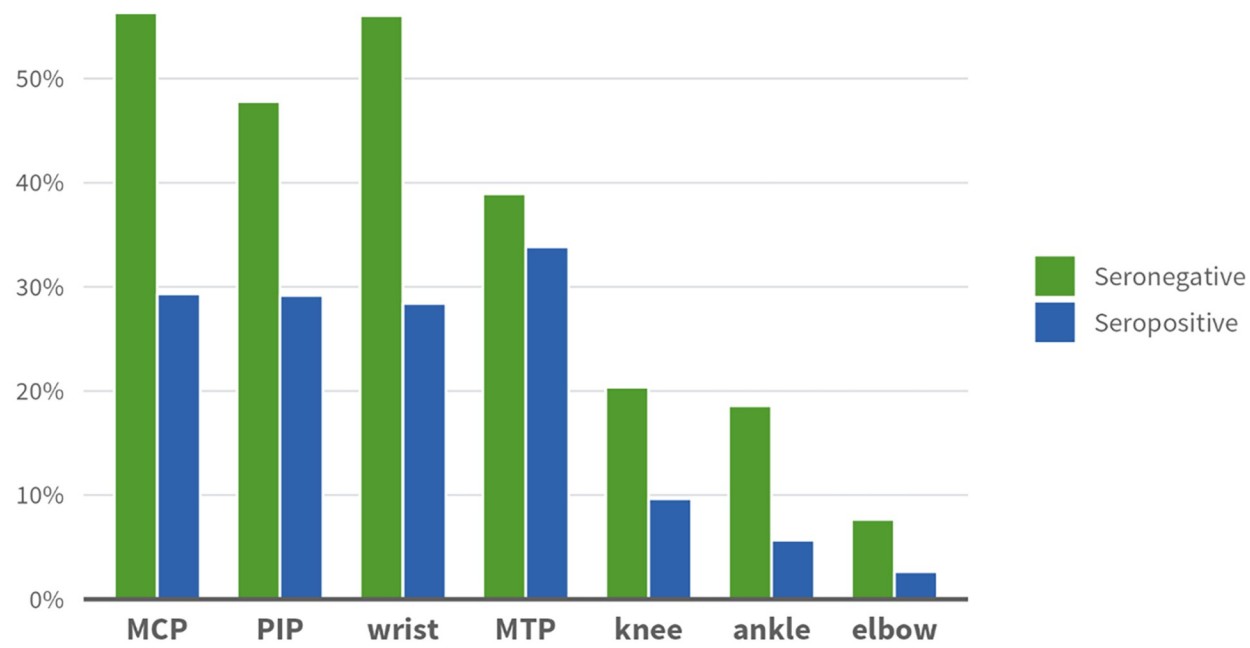

**Fig 1. Proportions of patients with symmetrical swelling by anatomical site in seropositive and seronegative patients, at the initial presentation of RA.**

fatigue, the corresponding numbers were 45 (18, 70), 40 (14, 69) and 52 (31, 73) (p<0.001, adjusted for age and sex) and for PGA they were 50 (25, 67), 48 (23, 64) and 53 (32, 70) (p<0.001, adjusted for age and sex) (Table 2).

The overall median (IQR) HAQ was 0.9 (0.5, 1.4) for all patients, 0.88 (0.38, 1.4) for seropositive patients and 1.1 (0.75, 1.5) for seronegative patients at the initial presentation (p<0.001, adjusted for age and sex) (Table 2).

## Initial medication

A total of 17% of seropositive patients and 23% of seronegative patients started with MTX monotherapy, 72% and 67% started MTX + csDMARD, 10% and 11% started csDMARD(s) only, while bDMARDs and JAK-inhibitors were started only for a few patients. GCs were started for 82% of seropositive and 94% of seronegative patients (Table 2).

## Effect of duration of symptoms on the clinical picture

Between patient groups by duration of symptoms, there were notable differences in terms of ACPA-positivity, DAS28, ESR, pain and HAQ. For other variables, such as SJC46, TJC46, CRP, Dr.global, fatigue and PGA, the differences between groups were minor and mostly statistically not significant (S2 Table).

*ACPA-positivity* was the most prevalent with a proportion of 77% in group 3, compared to 70% and 70% % in other groups (p = 0.045, adjusted for age and sex). There were similar results also for seropositivity (78%, 77% and 82% in groups 1,2 and 3, p = 0.115)- and RF-positivity (68%, 67% and 71% in groups 1, 2 and 3, p = 0.324) (S2 Table).

## Disease activity

The median (IQR) DAS28 was 4.1 (3.3, 5.1) in group 1, 4.0 (3.2, 4.8) in group 2 and 3.8 (3.1, 4.5) in group 3 (p = 0.016, adjusted for age and sex). The median (IQR) ESR was 27 (12, 44) in group 1, 20 (10, 36) in group 2 and 18 (9, 34) in group 3 (p<0.001, adjusted for age and sex) and the proportions of patients with a normal level of ESR were 55%, 61% and 66% in the same groups, respectively (p = 0.046, adjusted for age and sex) (S2 Table).

## PROs

For median (IQR) pain, the values in groups 1, 2 and 3 were 62 (35, 78), 52 (30, 72) and 50 (26, 70) (p<0.001, adjusted for age and sex). For median (IQR) HAQ, the values in the corresponding groups were 1.10 (0.50, 1.60), 0.9 (0.50, 1.40) and 0.75 (0.38, 1.20) (p = 0.002, adjusted for age and sex) (S2 Table).

## Clinical variables between ACPA-positive but RF-negative and RF-positive but ACPA-negative patients

A total of 163 (11% of seropositive patients) were only ACPA-positive and 118 (8% of seropositive patients) were only RF-positive. The mean (SD) age was 53 (17) for solely ACPA- and 62 (14) for solely RF-positive patients (p<0.001) and the duration of symptoms for the same groups were 6 (3, 13) and 4 (2, 7) (p = 0.222). The median (IQR) DAS28 was 3.6 (2.8, 4.2) for only ACPA-positive patients and 4.2 (3.5, 5.0) for only RF-positive patients (p<0.001). The same numbers for median (IQR) HAQ were 0.62 (0.3, 1.1) and 1.1 (0.5, 1.5) (p = 0.033). Besides DAS28 and HAQ, there were no statistically significant differences, though all of the clinical variables were higher for only RF-positive patients (Table 3).

## Discussion

Our main observation was that early RA is still mainly presented as symmetric arthritis, as three out of four patients had symmetrical swelling in any of the anatomical sites of the 1987 criteria at the initial presentation [1]. Overall, the wrists were the most commonly swollen joints by a significant margin (S1 Table). Symmetrical swelling was detected more often overall

**Table 3. Clinical data of patients that were only ACPA- or RF-positive.**

| Variable | Available data for ACPA-positive patients | ACPA-positive | Available data for RF-positive patients | RF-positive | p-value |
|---|---|---|---|---|---|
| **n, % of seropositive patients** | | 163 (11%) | | 118 (8%) | |
| **Age** | | 53 (17) | | 62 (14) | <0.001 |
| **Sex** | | 112 (69%) | | 84 (71%) | |
| **Duration of symptoms** | 124 (76%) | 6 (3, 13) | 87 (74%) | 4 (2, 7) | 0.222 |
| **Median (IQR) SJC46** | 156 (96%) | 4 (2, 8) | 113 (96%) | 6 (3, 10) | 0.193 |
| **Median (IQR) TJC46** | 156 (96%) | 4 (2, 7) | 113 (96%) | 7 (3, 12) | 0.056 |
| **Median (IQR) CRP** | 152 (93%) | 5 (3, 13) | 111 (94%) | 9 (3, 26) | 0.458 |
| **Median (IQR) ESR** | 137 (84%) | 13 (7, 28) | 99 (84%) | 20 (10, 35) | 0.780 |
| **Median (IQR) DAS28** | 148 (91%) | 3.6 (2.8, 4.2) | 108 (92%) | 4.2 (3.5, 5.0) | <0.001 |
| **Median (IQR) Dr.global** | 137 (84%) | 35 (25, 50) | 103 (87%) | 40 (27, 50) | 0.194 |
| **Median (IQR) PROs** <br> **Pain** <br> **Fatigue** <br> **PGA** | 146 (90%) <br> 131 (80%) <br> 146 (90%) | 50 (27, 75) <br> 50 (20, 70) <br> 50 (25, 64) | 104 (88%) <br> 91 (77%) <br> 103 (87%) | 58 (33, 77) <br> 46 (18, 74) <br> 51 (32, 70) | 0.145 <br> 0.767 <br> 0.402 |
| **Median (IQR) HAQ** | 134 (82%) | 0.62 (0.3, 1.1) | 93 (79%) | 1.1 (0.5, 1.5) | 0.033 |

and at all sites in ACPA negative patients. This was an expected finding since more swollen joints are required to fulfill the criteria for seronegative RA [5]. Seronegative patients had also higher SJC46 at the initial presentation (median 5 *versus* 10). Interestingly, the presence of symmetrical swelling was only slightly higher for seronegative patients in the MTPs (34% *versus* 39%, p = 0.097) (Table 2) despite higher SJC46.

## Clinical measures and medication

In terms of clinical data, seronegative RA presented significantly higher activity by almost all variables used in the study (Table 2). Similar differences were found between solely ACPA-positive and solely RF-positive patients, though most of the findings weren't statistically significant (Table 3). In ESR, the difference was only minor between seropositive and seronegative patients (20 *versus* 24, p = 0.172) (Table 2). Seropositive patients were more likely to fulfill the ACR/EULAR 2010 criteria (82% *versus* 78) at the time of diagnosis. Patients with seropositive RA were also current smokers more often, (21% *versus* 12% current smokers) (Table 1), as smoking is a known risk-factor for ACPA-positive RA [24]. In terms of medication, GCs and MTX as monotherapy was used more frequently in patients that were diagnosed with seronegative RA. The more prevalent use of GCs indicates that some of the diagnoses might have been actually polymyalgia rheumaticas, as our previous studies have suggested [10,11].

## Comparison to other studies

The results between seropositive and negative RA were somewhat contradictory compared to earlier research, a change likely caused by aiming at earlier diagnosis in accordance with the 2010 classification criteria. A French study of 354 patients diagnosed between 2002 and 2005 showed that seropositive patients had higher average DAS28 (5.3 *versus* 5.0) and HAQ (1.0 *versus* 0.9) than seronegative patients, but similar joint counts and VAS-values of pain [12]. In our study, the variables were significantly higher for seronegative RA (mean DAS28 of 4.8 *versus* 3.7 and 1.2 *versus* 0.93 for HAQ). However, the populations are not directly comparable, as the study had slightly different threshold values for ACPA and RF.

A Danish study of 198 newly diagnosed seropositive and 36 seronegative patients diagnosed in 2010 to 2013 showed lower median SJC44 (8 *versus* 17), Dr.global (39 *versus* 49) and DAS28 (3.4 *versus* 3.9), but similar PROs at the initial presentation in ACPA positive *versus* negative patients [25]. On the contrary, our population showed significantly lower PROs at the initial presentation in seropositive *versus* seronegative patients (51 *versus* 61 for pain, 40 *versus* 52 for fatigue and 48 *versus* 54 for PGA) (Table 2), in addition to other clinical variables.

In terms of duration of symptoms, a previous study of patients diagnosed in 2007–2012 in Canada showed a duration of symptoms of 6.5 months for patients with seropositive RA and 5.4 months for patients with seronegative RA [26]. This is in line with our results, where the median duration of symptoms was five months for seropositive and four months for seronegative RA (Table 1).

## The effect of duration of symptoms on the initial presentation

The overall median SJC46 was the same in all groups by duration of symptoms (median 6 joints) (S2 Table) and median DAS28 was higher in the group of patients with a delay of < 3 months compared to patients with a delay of >6 months (4.1 *versus* 3.8, p = 0.016, adjusted for age and sex). Interestingly, patients with a delay of <3 and 3–6 months were also older than in the group with a delay of >6 months (mean age of 61 and 58 in groups with a delay of <3 and 3–6 months and 56 in the group with a delay of >6 months, p<0.001) (S2 Table). It has been shown that older patients utilize health care services more frequently, which might explain the

finding [27]. The overall median CRP and ESR were higher in patients who were diagnosed earlier (11 in the group with a delay of <3 months, *versus* 8 and 7 in groups with a delay of 3–6 months and >6 months for CRP) and (27 *versus* 20 and 18 for ESR in the same groups, correspondingly). High laboratory values might have influenced towards early referrals, which in turn lowers the duration of symptoms. In addition, the median VAS-pain and HAQ were also significantly higher in patients who were diagnosed earlier (62 in the group with a delay of <3 months *versus* 52 and 50 in groups with 3–6 and >6 months, for pain) and (1.10 *versus* 0.90 and 0.75 for the same groups for HAQ), but ACPA was positive less frequently (77% in the group with a delay of >6 months *versus* 70% and 70% in other groups, p = 0.045) (S2 Table), which indicates that patients experiencing more severe symptoms and limitations in physical activity might seek health care services earlier. ACPA-positivity isn't necessarily associated with the severity of symptoms. A previous Polish study has shown that the second most important factor for the duration of symptoms of RA is patients' conviction that the condition will resolve on its own [28].

## Strengths and limitations

*The main strength* of this study was its large patient population from almost all health care regions, in Finland and that patient data was documented recently allowing an accurate presentation of the current clinical picture.

### Weaknesses

Although all health care regions are included in the quality register, at the time that was defined as the focus period, not all regions actively recorded clinical data. Therefore, probably not all patients with incident RA could be included. Furthermore, although not recommended, patients may still be diagnosed along with receiving treatment initiation in private practice, which are not involved in the quality register, yet.

One of the weaknesses in register studies is missing data, although the completeness was as high as ≥95% for joint counts and ≥78% for PROs.

## Conclusions

Our results indicate that the initial presentation of early RA is still mostly symmetric seropositive arthritis. Furthermore, we found that seronegative RA has significantly higher inflammatory activity and disease burden at the initial presentation by several clinical variables, compared to seropositive cases. However, 25% of patients in the current study didn't have symmetrical swelling at the initial presentation, which is why all patients with clinical joint inflammation of unknown reason ought to be tested for ACPA and RF and referred to a rheumatology unit if ACPA and/or RF ispositive. Our observations are encouraging in terms that patients experiencing more severe pain and decrease in functional ability might seek health care services earlier. We also found that ACPA—positivity is not necessarily associated with the severity of symptoms.

## Supporting information

**S1 Table. Proportions of patients with a swollen joint from each of the sites in SJC46.**
(DOCX)

**S2 Table. Clinical data of patients with incident RA by diagnostic delay.**
(DOCX)

## Author Contributions

**Data curation:** Henri Salo.

**Formal analysis:** Henri Salo.

**Investigation:** Tuulikki Sokka-Isler.

**Methodology:** Lauri Weman, Tuulikki Sokka-Isler.

**Software:** Henri Salo.

**Supervision:** Tuulikki Sokka-Isler.

**Visualization:** Lauri Weman.

**Writing – original draft:** Lauri Weman, Tuulikki Sokka-Isler.

**Writing – review & editing:** Henri Salo, Laura Kuusalo, Johanna Huhtakangas, Johanna Kärki, Paula Vähäsalo, Maria Backström, Tuulikki Sokka-Isler.

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
