## [Decision Letter · Decision Letter 0]

25 Jan 2023

PONE-D-22-32344Initial presentation of early rheumatoid arthritisPLOS ONE

Dear Dr. Weman,

Thank you for submitting your manuscript to PLOS ONE. After careful consideration, we feel that it has merit but does not fully meet PLOS ONE’s publication criteria as it currently stands. Therefore, we invite you to submit a revised version of the manuscript that addresses the points raised during the review process.

We look forward to receiving your revised manuscript.

Kind regards,

Sadiq Umar

Academic Editor

PLOS ONE

Journal Requirements:

   "We would like to thank The Finnish Psoriasis Association, The Finnish Society for Rheumatology,and Amgen. The author also acknowledges the Research Committee of the Kuopio University Hospital Catchment Area for the State Research Funding (project 5041730, Kuopio, Finland)."

  "NO"

   "NO"

7. Please ensure that you include a title page within your main document. You should list all authors and all affiliations as per our author instructions and clearly indicate the corresponding author.

8. We note that you have included the phrase “data not shown” in your manuscript. Unfortunately, this does not meet our data sharing requirements. PLOS does not permit references to inaccessible data. We require that authors provide all relevant data within the paper, Supporting Information files, or in an acceptable, public repository. Please add a citation to support this phrase or upload the data that corresponds with these findings to a stable repository (such as Figshare or Dryad) and provide and URLs, DOIs, or accession numbers that may be used to access these data. Or, if the data are not a core part of the research being presented in your study, we ask that you remove the phrase that refers to these data.

9. Please include your full ethics statement in the ‘Methods’ section of your manuscript file. In your statement, please include the full name of the IRB or ethics committee who approved or waived your study, as well as whether or not you obtained informed written or verbal consent. If consent was waived for your study, please include this information in your statement as well. 

10. Please include a copy of Table 3 and 4 which you refer to in your text on page 10 and 15.

11. Please include captions for your Supporting Information files at the end of your manuscript, and update any in-text citations to match accordingly. Please see our Supporting Information guidelines for more information: http://journals.plos.org/plosone/s/supporting-information. 

Reviewers' comments:

Reviewer's Responses to Questions

**Comments to the Author**

1. Is the manuscript technically sound, and do the data support the conclusions?

Reviewer #1: Yes

Reviewer #2: Yes

2. Has the statistical analysis been performed appropriately and rigorously? 

Reviewer #1: Yes

Reviewer #2: Yes

3. Have the authors made all data underlying the findings in their manuscript fully available?

Reviewer #1: Yes

Reviewer #2: No

4. Is the manuscript presented in an intelligible fashion and written in standard English?

Reviewer #1: Yes

Reviewer #2: Yes

5. Review Comments to the Author

Reviewer #1: The authors describe the initial presentation of early rheumatoid arthritis patients in Finland, using national databases. The result may be of interesting to readers, however, I have a couple of concerns. Specific comments are listed below:

1. The authors compare clinical items between ACPA-positive RA patients and negative patients, this is a bit confusing because so called seropositive RA includes RF+/ACPA- patients. This study may include such patients in ACPA negative RA patients, which shows a bit different nuance compared to 'seronegative' RA. More explanation of this may be needed to clarify the characteristics of patient group. I also be interested the clinical feature of ACPA-/RF+ RA patients (can be difficult to investigate, though).

2. We routinely use EULAR/ACR criteria for RA diagnosis in daily clinical practice, and due to the setting of scoring system, it is understandable ACPA negative (seronegative??) RA patients show more swollen joints (symmetrical) or high CRP levels. Therefore, it is natural to predict the results in this study. In other words, what is the most interesting findings in this study, compared to previous findings? Please address more clearly.

3. This study dose not describe initial therapy after diagnosis of RA. Are there any differences (if can)?

Reviewer #2: The authors studied the initial clinical presentation of alarge number of RA patients using national databases from Finland, with a particular interest in the impact of the presence of ACPA on the clinical presentation. A fair concern related to the objective of this study is that patients that do not present with symmetric polyarthritis could be having their diagnosis delayed. They concluded that incident RA presents mainly as symmetric arthritis, with higher disease severity in ACPA-negative patients. . This is not surprising, since as the authors themselves mentioned, the current classification criteria gives weight to seropositivity and, in negative patients, to the number of joints. On the other hand, it was surprising that ACAP-pos patients had a longer delay in diagnosis than ACPA-neg, when we know that ACPA is correlated with disease severity. For me, this suggests that there may be present a recruitment bias in this study: the ACPA-neg patients recruited could be representative of a more severe spectrum, while patients with milder forms are being left undiagnosed as RA, perhaps receiving other labels, such as "undifferentiated arthritis". The authors should discuss that possibility since it can have a great bearing on the main conclusions of the study. 

Other issues/recommendations: 

1. The authors should comment on the criteria for the request of ACPA, since about 11% of incident RA patients were not included in the analysis because they did not have ACPA available in their records. Could all these patients be RF positive, with high disease activity and, therefore, ACPA was forsaken because of obvious diagnosis? This could alter some of the conclusions that ACPA-neg patients have higher disease activity and shorter diagnostic delay.

2. The variables in Methods should be presented in full sentences, with better descriptions, and not by hyphenated-items. Particular confusing are the variables used to represent the duration of disease: "duration of symptoms", "fulfillment of ACR/EULAR 2010-criteria" and "diagnostic delay". These terms should be standardized throughout the text.

3. In terms of disease duration, the median (IQR) of 4 (2,10) months is very short, indicating that most (if not all) patients have relatively early (< 1y) disease. This should be discussed: is it true that very few patients in Finland are being diagnosed with longer than 1 year disease duration?  Perhaps the ranges of disease duration could be presented, particularly for the groups by diagnosis delay.

4. Tables 3 and 4 are supplementary - this should be indicated in the main text when they are referred to.

6. PLOS authors have the option to publish the peer review history of their article (what does this mean?). If published, this will include your full peer review and any attached files.

Reviewer #1: No

Reviewer #2: No

---

## [Author Response · Author response to Decision Letter 0]

24 Apr 2023

Dear Sadiq Umar

We have now made all the corrections you suggested to us for the manuscript called ''initial presentation of rheumatoid arthritis''. I hope this is now acceptable for your paper.

B.R Dr Weman

---

## [Decision Letter · Decision Letter 1]

12 Jun 2023

Initial presentation of early rheumatoid arthritis

PONE-D-22-32344R1

Dear Dr. Weman,

We’re pleased to inform you that your manuscript has been judged scientifically suitable for publication and will be formally accepted for publication once it meets all outstanding technical requirements.

Kind regards,

Sadiq Umar

Academic Editor

PLOS ONE

Additional Editor Comments (optional):

The Manuscript looks good after revision though need some minor changes as suggested by Reviewer 3.

Reviewers' comments:

Reviewer's Responses to Questions

**Comments to the Author**

1. If the authors have adequately addressed your comments raised in a previous round of review and you feel that this manuscript is now acceptable for publication, you may indicate that here to bypass the “Comments to the Author” section, enter your conflict of interest statement in the “Confidential to Editor” section, and submit your "Accept" recommendation.

Reviewer #1: All comments have been addressed

Reviewer #2: All comments have been addressed

Reviewer #3: (No Response)

2. Is the manuscript technically sound, and do the data support the conclusions?

Reviewer #1: Yes

Reviewer #2: Yes

Reviewer #3: Yes

3. Has the statistical analysis been performed appropriately and rigorously? 

Reviewer #1: Yes

Reviewer #2: Yes

Reviewer #3: N/A

4. Have the authors made all data underlying the findings in their manuscript fully available?

Reviewer #1: Yes

Reviewer #2: Yes

Reviewer #3: Yes

5. Is the manuscript presented in an intelligible fashion and written in standard English?

Reviewer #1: Yes

Reviewer #2: Yes

Reviewer #3: Yes

6. Review Comments to the Author

Reviewer #1: The revised manuscript shows substantial improvement. The authors adequately answered to my concerns.

Reviewer #2: (No Response)

Reviewer #3: The manuscript has been written well, but I have some minor suggestions: (Please look at comments on pdf form). I suggest talking about the relation between cardiovascular risk factors and metabolic diseases with early rheumatoid arthritis in your manuscript.

---

## [Editor Report · Acceptance letter]

19 Jun 2023

PONE-D-22-32344R1 

Initial presentation of early rheumatoid arthritis 

Dear Dr. Weman:

I'm pleased to inform you that your manuscript has been deemed suitable for publication in PLOS ONE. Congratulations! Your manuscript is now with our production department. 

Kind regards, 

on behalf of

Dr. Sadiq Umar 

Academic Editor

PLOS ONE